# HIF-1α-mediated feedback prevents TOR signalling from depleting oxygen supply and triggering stress during normal development

Yifan Zhao[1], Cyrille Alexandre[1], Gavin Kelly [1], Jean-Paul Vincent [1,2] & Gantas Perez-Mockus [1,2]

Growth deceleration before growth termination is a universal feature of growth during development. Transcriptomics analysis reveals that during their two-day period of growth deceleration, wing imaginal discs of *Drosophila* undergo a progressive metabolic shift from oxidative phosphorylation towards glycolysis. Ultra-sensitive reporters of HIF-1α stability and activity show that imaginal discs become increasingly hypoxic during development in normoxic conditions, suggesting that limiting oxygen supply could underlie growth deceleration. We confirm the expectation that rising levels of HIF-1α dampen TOR signalling activity through transcriptional activation of REDD1. Conversely, excess TOR leads, in a tissue-size-dependent manner, to hypoxia, which boosts HIF-1α levels and activity. Thus, HIF-1α mediates a negative feedback loop whereby TOR signalling triggers hypoxia, which in turn reduces TOR signalling. Abrogation of this feedback by *Sima/HIF-1α* knockdown leads to cellular stress, which is alleviated by reduced TOR signalling or a modest increase in environmental oxygen. We conclude that Sima/HIF-1α prevents TOR-mediated growth from depleting local oxygen supplies during normal development.

In many instances, growth during development follows an S-shaped curve, with a rapid initial phase followed by an inflexion point and a progressive slowing down. Examples of such growth behaviour have been documented in whole animals, including mammals[1–3], birds[4], crustaceans[5,6], insects[7], and planarian worms[8], as well as in organs and appendages such as miniature swine livers[9], human tibias[10], bird wings[11] and *Drosophila* eye precursors[12]. Natural populations follow a similar growth pattern, which can be modelled by logistic functions[13]. In this case, growth deceleration can be attributed to limited resources[14]. However, in a well-fed animal, resources are not expected to be limiting, and the basis of growth deceleration is not known.

In multicellular organisms, growth is controlled by a wide range of systemic and intrinsic signals. Systemic cues include growth hormones and insulin-like growth factors that coordinate the growth of multiple organs[15], and modulate final size according to nutrient availability[15–18]. For example, in *Drosophila*, a pulse of the steroid hormone ecdysone has been shown to trigger proliferation arrest at the time of pupariation[19,20]. However, despite the importance of ecdysone signalling for appendage growth, there is no documented evidence that hormonal signalling declines during development. On the contrary, ecdysone titres progressively rise before the pulse that triggers proliferation arrest. Therefore, there is no simple correlation between ecdysone signalling and growth deceleration. Tissue-intrinsic cues

[1]The Francis Crick Institute, London, UK. [2]These authors contributed equally: Jean-Paul Vincent, Gantas Perez-Mockus. e-mail: jp.vincent@crick.ac.uk; gantas.perezmockus@crick.ac.uk

from morphogen signalling are also known to contribute to growth control both in *Drosophila* imaginal discs and other tissues[21–23]. However, as with hormonal signalling, morphogen signalling continues to rise, at least in *Drosophila* wing imaginal discs, suggesting that it cannot account for growth deceleration. Finally, theoretical considerations have suggested that mechanical feedback could dampen growth as tissue size increases[24–26]. However, so far, there is no experimental evidence that mechanical constraints govern whole organ growth deceleration. Therefore, despite the universal nature of growth deceleration, its molecular basis remains unknown.

To study growth deceleration, we chose wing imaginal discs of *Drosophila*, a well-established model of growth control[21,22,24,27]. Wing discs originate from a group of ~30 cells set aside during embryogenesis[28–30]. Over the course of the three larval instars (L1-L3), they undergo 10-11 cell cycles to generate the 30,000–50,000 cells that make up adult wings[31,32]. Volume measurements confirmed that during the third larval instar (L3), wing disc growth decelerates. Bulk RNA-Seq at specific time points during this period revealed that mRNAs encoding components of aerobic metabolism become increasingly less abundant with disc age, suggesting that discs may become mildly hypoxic even when raised in normoxic conditions. In all metazoans, the response to environmental hypoxia is mediated by Hypoxia Inducible Factor-1α (HIF-1α)[33], a transcription factor known as Similar (Sima) in *Drosophila*[34]. For consistency, we will use the mammalian name, HIF-1α, throughout this paper. Under normoxia, *Drosophila* HIF prolyl hydroxylase (Hph, AKA Fatiga; PHD in mammals) hydrolyses HIF-1α at specific proline residues within the so-called oxygen-dependent degradation domain (ODD)[35–39], thus earmarking it for ubiquitylation by the von Hippel Lindau (VHL) protein and degradation by the proteasome[40–42]. Under hypoxia, HIF-1α hydroxylation and hence ubiquitylation are reduced[37], leading to its accumulation in the nucleus and activation of target genes[43]. We developed sensitive reporters of hypoxia and of HIF-1α activity to confirm that the classic hypoxia response is activated during development in normoxia. With these reporters, we show that rising HIF-1α activity during development correlates with decreasing TOR signalling, in accordance with prior evidence that HIF-1α represses Target of Rapamycin (TOR) activity. Unexpectedly, excess TOR activity leads to increased hypoxia and hence HIF-1α activity, which in turn suppresses TOR activity. This HIF-1α-mediated negative feedback prevents growth from putting pressure on oxygen supplies, perhaps accounting for growth deceleration. Abrogation of this feedback by inactivation of HIF-1α leads to cellular stress, demonstrating the importance of reining in TOR signalling.

## Results

### Transcriptomic profiling of L3 wing discs suggests a progressive decrease in aerobic metabolism

We measured the volume (a proxy for tissue mass) of precisely timed wing discs, from the onset of L3, at 72 hours after egg laying (h AEL) until pupariation (Fig. 1a, b). Wing disc volumes, expressed as a ratio to starting volume, were plotted over time (Fig. 1c), and the relative growth rate was determined as its first derivative, normalised to the instantaneous volume. This analysis confirmed that the relative growth rate progressively decelerates, declining from 9% per hour at the beginning of L3 to 2% per hour at the end of L3, before pupariation (Fig. 1d).

We then sought to identify mRNAs that correlate or anti-correlate with the instantaneous growth rate. Wing discs grow relatively homogeneously[44], alleviating the need for spatially resolved analysis. Bulk RNA-Seq was performed on precisely timed L3 wing discs at seven developmental time points, ranging from 80 to 118 h AEL. For each expressed gene, a time-course expression profile was generated, and the Pearson correlation coefficient ($r$) with growth rate was obtained. Genes with an absolute $|\pm r|$ value exceeding 0.8 were considered of

interest. They were analysed using the Kyoto's Encyclopedia of Genes and Genomes (KEGG) database resource[45] to reveal the molecular functions of mRNAs that correlate or anti-correlate with growth rate. Two groups of downregulated genes stood out (Fig. 1e). Group I encompasses genes associated with DNA, RNA and protein synthesis, activities that are required for biomass accumulation. The presence of these genes among our hits is expected from growth deceleration; they will therefore not be considered further. Group II comprises genes involved in oxidative phosphorylation (OXPHOS), tricarboxylic acid (TCA) cycle and acetyl coenzyme A (acetyl-CoA) biosynthesis (Fig. 1f and Supplementary Fig. 1a–c), suggesting a progressive decrease in aerobic metabolism as wing discs grow towards their final size. In addition, our analysis revealed a significant increase in the expression of many genes encoding glycolytic enzymes, including *phosphofructokinase* (*Pfk*), *aldolase 1* (*Ald1*), *triose-phosphate isomerase* (*Tpi*), *glyceraldehyde-3-phosphate dehydrogenase 2* (*Gapdh2*) and *phosphoglycerate kinase* (*Pgk*) (Supplementary Fig. 1d). Of note, mRNA encoding *Glucose transporter 1* (*Glut1*) was found to increase 14-fold during L3, suggesting an increase in anaerobic metabolism. A decrease in OXPHOS combined with an increase in anaerobic metabolism is reminiscent of the response to hypoxia[46–52]. Several HIF-1α target genes were indeed found to be upregulated, and expression of an enhancer trap[53,54] inserted in the gene encoding *lactate dehydrogenase* (*Ldh*), a classic hypoxia response gene[55] increased, albeit mildly (Supplementary Fig. 1e, e′). We next set out to design dedicated sensors to assess the level of hypoxia in wing discs.

### Developing L3 wing discs become increasingly hypoxic in environmental normoxia

A previously described transgenic oxygen sensor termed nlsTimer takes advantage of a fluorophore that matures to two distinct isoforms, with a green variant being favoured at low oxygen tension and a red one at high oxygen tension[56] (Supplementary Fig. 2a). Thus, the red-to-green ratio after a pulse of expression gives an indication of oxygen levels. We triggered expression of this sensor in whole larvae at 72 and 84 h AEL and performed ratiometric fluorescence analysis 32 hours later (the time needed for maturation of the fluorophores) (Supplementary Fig. 2b). The results suggest that whole larva oxygen levels could indeed decrease as development proceeds (Supplementary Fig. 2c–c″). Unfortunately, it was not possible to estimate cellular oxygen specifically in wing discs because the sensor had an adverse effect on growth (Supplementary Fig. 2d). Therefore, the activity of nlsTimer is suggestive of mild larva-wide hypoxia in late L3, but one cannot infer whether this is reflected in reduced oxygen in wing discs.

Another means of estimating oxygen levels is by leveraging the mechanism whereby oxygen tension modulates the stability of HIF-1α[33]. The ODD domain of HIF-1α confers oxygen-modulated degradation to heterologous proteins (e.g. GFP), and this is the basis of another oxygen biosensor, GFP-ODD[57]. However, when assessing ODD-GFP protein stability (normalised to the mRFP reference protein expressed from a co-transgene), we detected no significant difference between early and late L3 discs (Supplementary Fig. 2e, e′), perhaps because of the limited sensitivity of the GFP-ODD reporter (only 1.5-fold stability increase in embryos shifted from normoxia to 5% environmental oxygen[57]). Recent work with mouse monocyte cells suggests that more sensitive measurements of hypoxia could be achieved with full-length HIF-1α[58]. We therefore created a reporter based on full-length *Drosophila* HIF-1α, dually tagged with V5 and neonGreen (mNeonGreen) and mutated to inactivate its interaction with HIF-1β, thereby preventing activation of hypoxia signalling by the reporter (Fig. 2a). DNA encoding this polypeptide (lacking native UTRs to prevent HIF-1α-specific translation regulation) was inserted downstream of a *ubiquitin* promoter (*ubi*), alongside a *ubi-mCherry* module for normalisation. In transgenic wing discs expressing this reporter, which we call FlHypox (Fly-Hypoxia), both V5 immunoreactivity and mNeonGreen

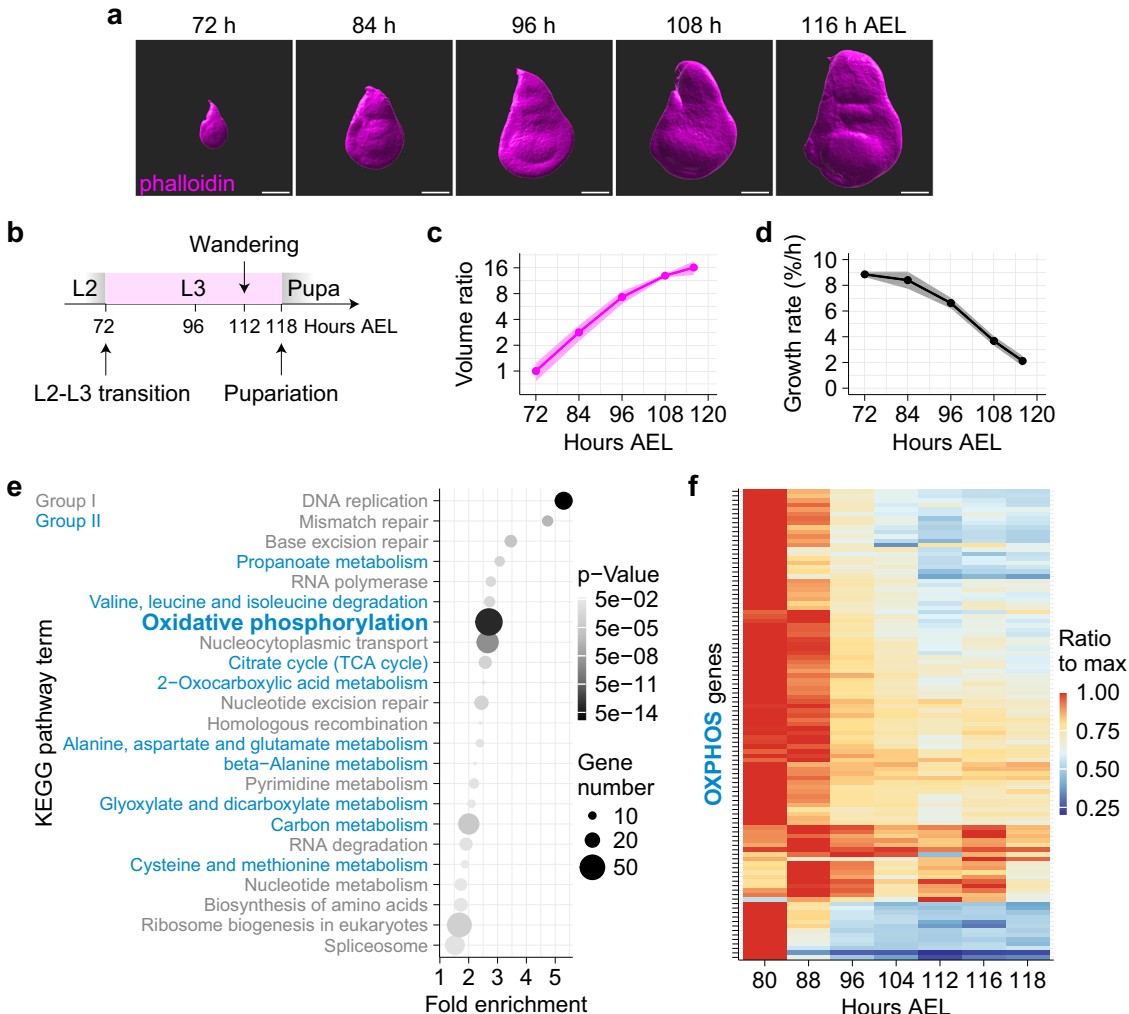

**Fig. 1 | Transcriptomic analysis of L3 wing discs suggests a progressive decrease in aerobic metabolism. a** Representative image of phalloidin-stained L3 wing discs used for volumetric reconstruction. Confocal images were taken and reconstructed under identical conditions. Scale bar = 100 μm. **b** Schematic timeline of larval development during L3 (72 to 118 h AEL). **c** Plot of disc volume, shown as dimensionless volumes (normalised to the mean initial volume). $n = 6$ for each time point, except for data at 84 h AEL, where $n = 5$. The error ribbon represents the standard deviation. Source data are provided as an Excel sheet. **d** Plot of relative growth rate, calculated as the time derivative of volume ($dV/dt$) divided by the instantaneous volume ($V_t$), where $V_t$ is a continuous function of time. The error ribbon represents the standard deviation estimated using bootstrap resampling with replacement.

Source data are provided as an Excel sheet. **e** KEGG pathway analysis of genes with expression profiles correlating positively with the growth rate (Pearson correlation coefficient $r > 0.8$). Group I (genes involved in the central dogma) and Group II (genes involved in aerobic respiration) are downregulated over time, with DNA replication and OXPHOS being the most significantly enriched. Source data are provided as an Excel sheet. **f** Transcriptional profile of OXPHOS genes during L3. For each gene, the expression level at each time point is normalised to the maximum value observed during L3. The normalised values are colour-coded according to the lookup table on the right. $n = 3$ biological replicates for each time point, except for data at 80 h AEL where $n = 2$.

fluorescence increased within 1 hour of transfer from normoxia (20.9%) to 5% environmental oxygen, although the increase was more apparent with V5 than mNeonGreen (Fig. 2b, b' and Supplementary Fig. 3). To confirm that FlHypox reflects hypoxia-induced HIF-1α stabilisation, we created a control *ubi*-driven transgene expressing an identical construct except for a point mutation in the key proline residue required for hydroxylation-dependent degradation. The resulting polypeptide (FlHypox[P850A]) was present at equally high levels in both 20.9% and 5% environmental oxygen (Fig. 2b, b' and Supplementary Fig. 3), as expected since oxygen-dependent degradation was abrogated. We therefore conclude that the increased FlHypox signal at 5% environmental oxygen is a direct consequence of reduced cellular oxygen. As an alternative means of estimating oxygen levels, we assessed the performance of Hypoxyprobe™ (AKA pimonidazole hydrochloride), which forms stable adducts with thiol groups in proteins[59] at low oxygen tension[60,61]. The resulting signal also increased

in hypoxic wing discs but not as markedly as the one from FlHypox (Fig. 2c). We conclude that FlHypox is the most effective means so far of directly assessing hypoxia in vivo.

As part of our effort to characterise oxygen levels in developing imaginal discs, we also devised means of estimating HIF-1α activity. Several reporters comprising consensus HIF-1α binding sites upstream of a minimal promoter and a marker gene have been generated[39,62–66]. For example, a LacZ reporter based on a murine 233-bp hypoxia-responsive enhancer comprising two consensus hypoxia response elements (HREs, 5'-(A/G)CGTG-3') and a cAMP response element was found to be activated at 5% environmental oxygen in transgenic *Drosophila* embryos[39]. However, this reporter showed no activity in milder hypoxic conditions (e.g., 11% O₂)[67], suggesting that improvements in sensitivity are needed. We reasoned that this could be achieved with a native hypoxia-responsive enhancer and by taking advantage of recent improvements in reporter gene design in *Drosophila*[20]. As a hypoxia-

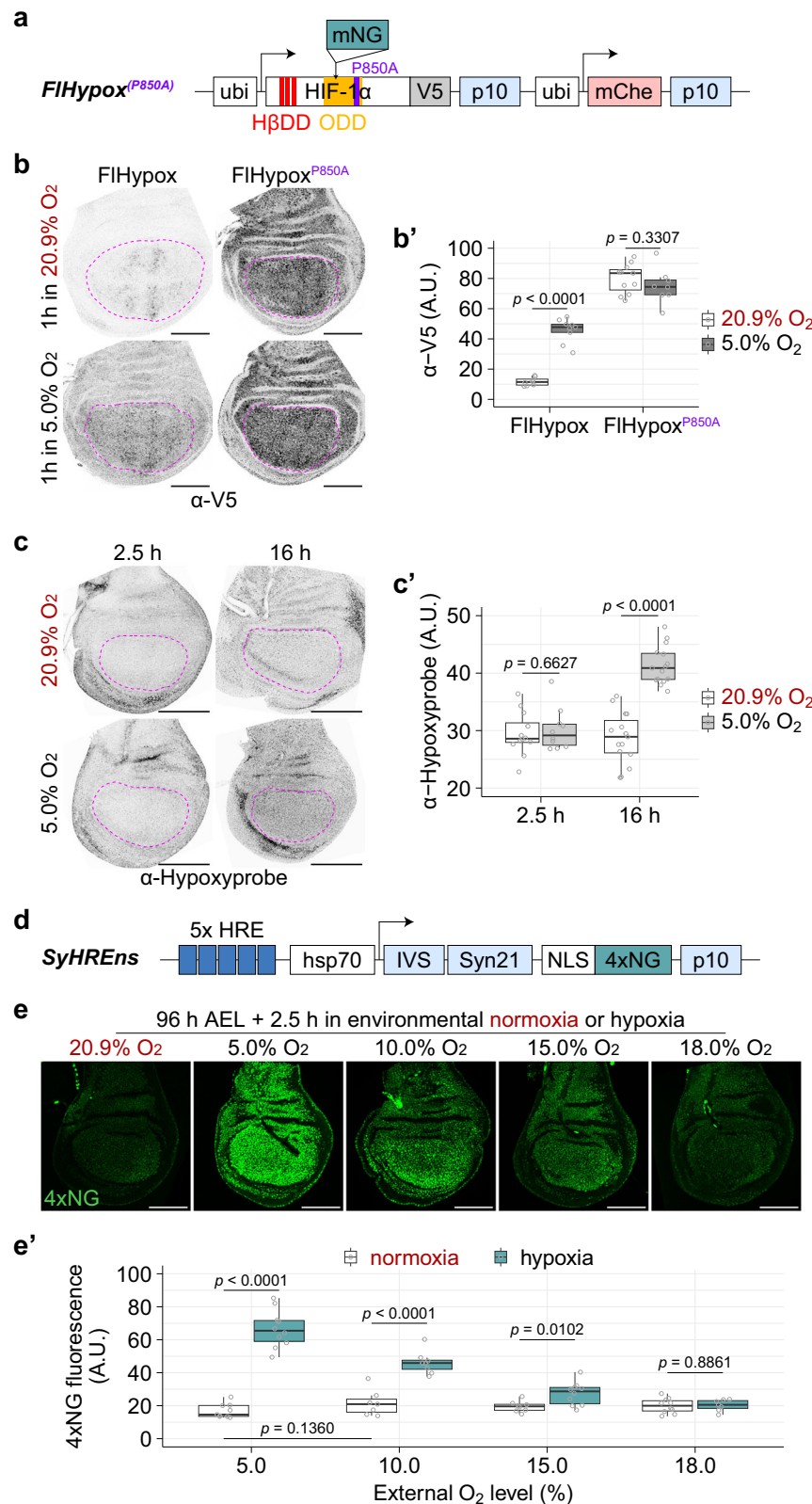

responsive enhancer, we used an 81-bp fragment from the *Drosophila hph* (AKA *fatiga*) gene comprising two putative HREs−HRE2 and HRE3−, which were characterised in S2 cells by Wappner and colleagues[68]. Five tandem copies of this fragment were inserted upstream of a minimal heat shock promoter driving the production of mRNA encoding nuclear-targeted NeonGreen tetramers (NLS4xNG). Viral untranslated sequences (syn21 at the 5' and p10 at the 3') were included to boost expression[69] (Fig. 2d). To assess the sensitivity and dynamic range of the resulting construct, which we call SyHREns (Synthetic HRE nuclear sensor), transgenic larvae from mid-L3 (96 h AEL) were cultured for 2.5 hours in different levels of oxygen, and the wing discs were then dissected out and imaged by fluorescence microscopy. The result showed that hypoxia activates the reporter in a severity-dependent manner (Fig. 2e, e'). A plot of reporter activity at

**Fig. 2 | Validation of hypoxia sensors under environmental hypoxia.**
**a** Schematic representation of the *FlHypox*- and *FlHypox*[P850A]-expressing transgenes. HβDD (HIF-1β dimerisation domain) was mutated at three residues (Q339E, V413E and Y417T) to prevent dimerisation with HIF-1β; ODD oxygen-dependent degradation domain. mNeonGreen was inserted between amino acid residues S725 and A726. The P850A mutation prevents hydroxylation of HIF-1α, thereby inhibiting oxygen-dependent degradation. **b-b'** Anti-V5 immunofluorescence analysis of L3 discs dissected from larvae cultured in either 20.9% (normoxia) or 5.0% environmental oxygen. All discs were imaged under identical conditions. Scale bar = 100 μm. Here, and in subsequent relevant images, the pouch was outlined with a dotted line to mark the area for quantification shown in (**b'**). $n \geq 8$ for each condition. Data normality was confirmed with the Shapiro–Wilk test. Statistical significance was assessed by a two-sided unpaired *t* test ($p \geq 0.05$, not significant). For a detailed description of box plots, see Supplementary Fig. 1e–e'. **c-c'** Hypoxyprobe™ adduct analysis of L3 wing discs dissected from larvae cultured in different environmental oxygen. All the discs shown in (**c**) were imaged under identical conditions. Quantification of wing pouch fluorescence intensity is shown in (**c'**). Scale bar = 100 μm. $n \geq 10$ for each condition. After confirming normality of the data, statistical significance was assessed by a two-sided unpaired *t* test. **d** Schematic representation of the *SyHREns* transgene. *HRE* hypoxia response element. Note that the presence of two copies of *SyHREns* leads to a mild developmental delay (approximately one hour over the approximately five days to pupariation) but has no other detectable effects on development. **e-e'** Fluorescence from SyHREns in L3 discs dissected from larvae cultured in different environmental oxygen for 2.5 h. Representative images are shown in (**e**) (all discs imaged under identical conditions), and quantification of wing pouch fluorescence intensity is shown in (**e'**). Scale bar = 100 μm. $n \geq 7$ for each time point. After confirming normality of the data, statistical significance was assessed by a two-sided unpaired *t* test.

various oxygen levels (normalised to that in normoxia) suggests that a relatively mild departure from normoxia (approximately 17% O₂) can be detected and that the response is relatively linear, with a 4-fold signal increase at 5% environmental oxygen (Supplementary Fig. 4a). These observations suggest that SyHREns is highly sensitive with an excellent dynamic range.

With sensitive assays in our hands, we proceed to characterise how oxygen levels and HIF-1α activity evolve during imaginal disc growth. Measurements of SyHREns activity from early to late L3 showed that HIF-1α activity gradually increases during this period, both in wing (Fig. 3a, a') and eye discs (Supplementary Fig. 4b, b'). In contrast, the activity of a control construct with mutated HREs (SyHREns[mut]) (Supplementary Fig. 4c) remained low and unchanged during L3 (Supplementary Fig. 4d). As a first test to determine if the rise in HIF-1α activity is due to hypoxia, wild-type larvae were fed Hypoxyprobe™ for 24 hours before dissection and fixation of wing discs at 96 or 116 h AEL. Immunofluorescence revealed a higher density of Hypoxyprobe™ adducts at the later stage (Fig. 3b, b'), consistent with increasing hypoxia. The signal from FlHypox also increased during L3, as shown with V5 immunoreactivity (Fig. 3c, c') or mNeonGreen fluorescence (Supplementary Fig. 5a). In contrast, the signal from FlHypox[P850A], which is insensitive to oxygen levels, decreased somewhat during this period (Fig. 3d, d' and Supplementary Fig. 5b). This could be due to a general decrease in translation rate in late L3, as suggested by a reduced rate of O-propargyl-puromycin incorporation (Supplementary Fig. 5c, c'). Therefore, increased FlHypox in late discs is particularly significant since it can be detected despite reduced translation. We conclude that wing discs become increasingly hypoxic as they grow during L3.

### HIF-1α dampens TOR signalling through activation of REDD1
In hypoxic cells, HIF-1α activity dampens growth by activating the expression of REDD1 (known as Scylla in *Drosophila*), which in turn suppresses the activity of TOR[70–72], a protein that integrates multiple inputs of anabolic metabolism regulation[73,74]. The same regulatory interactions could be deployed during development in normoxia, as evidenced by an approximately 5-fold increase in REDD1 expression, concurrent with decreasing TOR signalling activity (assayed by phosphorylated ribosomal protein S6 (pS6) immunoreactivity) during L3[75,76] (Supplementary Fig. 6a, b). This suppression of TOR activity is in accordance with the diminishing rate of O-propargyl-puromycin incorporation shown above (Supplementary Fig. 5c, c'). To assess whether rising HIF-1α activity contributes to decreased TOR activity, we performed gain and loss-of-function analysis. Western blot analysis revealed that, upon whole-disc knockdown of *HIF-1α* or *REDD1*, pS6 immunoreactivity (normalised to β-Tubulin or total S6) is higher than normal at the end of larval development (Supplementary Fig. 6c). Similarly, P compartment-specific knockdown of *HIF-1α* or *REDD1* also led to increased pS6 immunoreactivity relative to the control A

compartment (Fig. 3d, d', third and fourth columns). Conversely, *HIF-1α* overexpression suppressed S6 phosphorylation (Fig. 3d, d', second column). This effect of HIF-1α is likely mediated by transcriptional activation of REDD1 since overexpressed HIF-1α leads to a substantial increase in *REDD1* transcripts. Moreover, whole-disc knockdown of *REDD1* led to a 12% enlargement of adult wings (Supplementary Fig. 6f). We therefore suggest that during normal disc growth—as in environmental normoxia—rising HIF-1α activates REDD1, which in turn dampens growth by down-regulating TOR activity.

### Excess TOR activity leads to hypoxia in a tissue-size-dependent manner
While the inhibitory effect of HIF-1α on TOR activity was expected, we also found, as described below, that excess TOR signalling conversely induces mild hypoxia. This was first evidenced by a marked increase in SyHREns fluorescence intensity upon overexpression of *Ras homologue enriched in brain* (*Rheb*) with *UAS-Rheb* and *hh-GAL4* (Fig. 4a, a'), indicating that TOR signalling boosts HIF-1α activity, possibly through classic hypoxia-induced stabilisation. However, one cannot exclude an oxygen-independent effect, since TOR signalling has been shown to boost HIF-1α activity independently of stabilisation in HEK293 cells[77]. Moreover, an increase in global translation activity could also potentially contribute to a rise in HIF-1α protein level and hence increased SyHREns activity. To specifically test whether *UAS-Rheb* causes hypoxia, we assessed its effects on FlHypox and FlHypox[P850A] in the P compartment, using the A compartment as an internal control. Rheb overexpression increased the P/A ratio of V5 immunoreactivity for both constructs, albeit significantly more with FlHypox than FlHypox[P850A] (1.61 vs 1.36; Fig. 4b, b', third and fourth columns). The A/P ratio for the *ubi-mCherry* module also increased in response to Rheb overexpression, but also not as much as that from FlHypox (1.20 vs 1.61; Supplementary Fig. 7a, a', third column). We suggest that: the rise in the signal from *ubi-mCherry* reflects increased translation, the rise in FlHypox[P850A] is due to a combination of increased translation and a hypoxia-independent effect of TOR signalling on *HIF-1α* activity (possibly through HIF-1α's TOR signalling motif[77]), and the additional rise in FlHypox is due to hypoxia. We conclude that TOR activity leads to depletion of intracellular oxygen, and this was also confirmed with the HIF-1α-independent hypoxia marker Hypoxyprobe™ (Supplementary Fig. 7b, b')

As shown in Fig. 3 and Supplementary Fig. 5, the hypoxic state of imaginal discs rises progressively with age, suggesting that oxygen could be more limiting in old (large) than in young (small) discs. To investigate this possibility, we compared the effect of excess TOR activity on SyHREns activity in mid and late L3 discs. Quantification of the P/A ratio of SyHREns fluorescence showed that overexpressed Rheb has a much more marked effect on HIF-1α activity at late L3 than at mid L3 (Fig. 4c, c'). We conclude that the pressure on oxygen supply rises with tissue age/size. This may

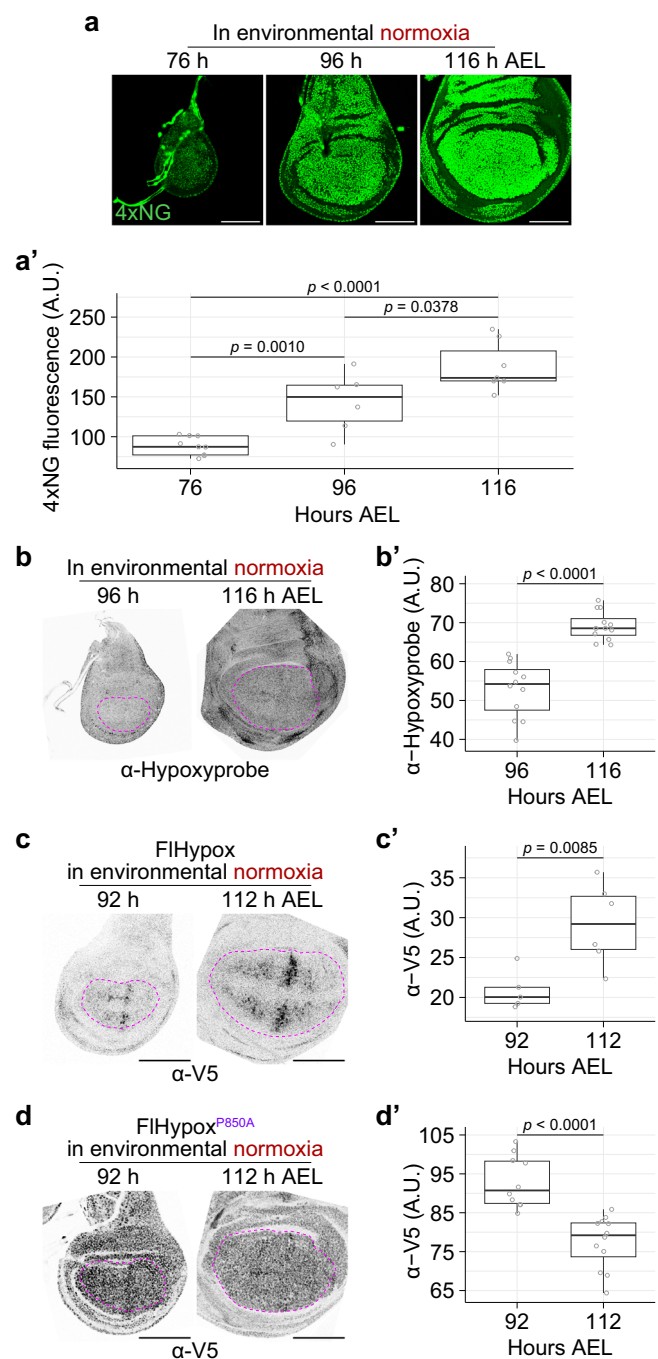

**Fig. 3 | Developing L3 wing discs become increasingly hypoxic in environmental normoxia. a-a'** Fluorescence from SyHREns in L3 discs increases during development in environmental normoxia. Representative images are shown in (**a**) (all discs imaged under identical conditions), and quantification of wing pouch fluorescence intensity is shown in (**a'**). Scale bar = 100 μm. $n \geq 7$ for each time point, except for data at 96 h AEL where $n = 6$. After confirming normality of the data, statistical significance was assessed by a two-sided unpaired $t$ test. **b-b'** Fluorescence from Hypoxyprobe™ adducts in L3 discs increases during development in environmental normoxia. 72 and 92 h AEL larvae were simultaneously reared on fly food containing 2 mg/mL Hypoxyprobe™ for 24 h prior to fixation. Representative images are shown in (**b**) (all discs imaged under identical conditions), and quantification of wing pouch fluorescence intensity is shown in (**b'**). Scale bar = 100 μm. $n = 12$ for each time point. After confirming normality of the data, statistical significance was assessed by a two-sided unpaired $t$ test. **c-c'** Fluorescence from anti-V5 staining in L3 discs expressing FlHypox increases during development in normoxia. Representative images are shown in (**c**) (all discs imaged under identical conditions) and quantification of wing pouch fluorescence intensity in (**c'**). Scale bar = 100 μm. $n \geq 5$ for each time point. After confirming normality of the data, statistical significance was assessed by a two-sided unpaired $t$ test. **d-d'** Fluorescence from anti-V5 staining in L3 discs expressing FlHypox$^{P850A}$ decreases during development under environmental normoxia. Representative images are shown in (**d**) (all discs imaged under identical conditions) and quantification of wing pouch fluorescence intensity is shown in (**d'**). Scale bar = 100 μm. $n \geq 10$ for each time point. After confirming normality of the data, statistical significance was assessed by a two-sided unpaired $t$ test.

explain why oxygen supplementation is required for the growth of cultured imaginal discs[78].

## Excess TOR activity leads to cellular stress in the absence of HIF-1α

Our results so far suggest that TOR signalling triggers mild hypoxia, which in turn limits TOR signalling. Reactive oxygen species (ROS), as assayed with Amplex Red, have been reported to increase in Sima/HIF-1α mutant tissue[51,79,80]. However, different assays using dichlorodihydrofluorescein diacetate (DCFH-DA) or Dihydroethidium have shown ROS to decrease upon *HIF-1α* knockdown[80,81]. Therefore, the role of HIF-1α in modulating ROS remains to be further elucidated, and we have turned to a different assay for cellular stress. We found that *HIF-1α* knockdown (e.g., in the P compartment) leads to activation of c-Jun N-terminal kinase (JNK) signalling (Fig. 5a, a', second

column), as indicated by the activity of TRE-NLS4xNG, a reporter of c-Jun N-terminal kinase (JNK) signalling activity (Supplementary Fig. 8a). This finding suggests that HIF-1α is required to prevent stress in normoxia. JNK signalling was further enhanced by concomitant *Rheb* overexpression (Fig. 5a, a', third and fourth columns), consistent with the notion that HIF-1α could prevent stress by dampening TOR activity. Accordingly, the mild stress caused by *HIF-1α* knockdown was fully rescued by inhibiting TOR activity (Supplementary Fig. 8b, b'). The role of HIF-1α in reining in TOR signalling was further investigated in *HIF-1α* mutant animals. A mutant allele, *sima/HIF-1α$^{KG07607}$* had previously been generated[82]. In addition, we generated a conditional allele, *HIF-1α$^{>E2-6>}$*, which can be converted by FLP recombinase to amorphic *HIF-1α$^{ΔE2-6}$* (Supplementary Fig. 8c). Both *sima/HIF-1α$^{KG07607}$* and *HIF-1α$^{ΔE2-6}$* were viable over a deficiency, albeit with a phenotype characterised by wing crumpling, loss of trichomes, and altered trichome shape (Supplementary Fig. 8d, e). Approximately 75% of homozygous *HIF-1α$^{ΔE2-6}$* adults had at least one crumpled wing. Reduced trichome density (a proxy for cell density[83]) was observed at the surface of non-crumpled wings (Supplementary Fig. 8e, e'), suggesting an increase in cell size, possibly as a result of excess TOR signalling. The crumpling phenotype was partially rescued by the addition of rapamycin in the larval food (Fig. 5b), confirming that it is caused by excess TOR activity, possibly by reducing oxygen availability and triggering cellular stress. To test the latter suggestion, we sought to test if the crumpling phenotype could be alleviated by increasing oxygen tension. We found 25% environmental oxygen to be toxic, even in wild-type animals (Supplementary Fig. 9a, a'). However, 22.5% environmental oxygen was well tolerated, and this mild departure from normoxia was sufficient to rescue Rheb-induced stress (Fig. 5c, c'). Conversely, mild environmental hypoxia enhanced Rheb-induced stress (Supplementary Fig. 9b, b'). These findings suggest that the TOR-induced cellular stress is due to a mismatch between oxygen demand and availability.

## Discussion

In this paper, we have shown that, in normoxic conditions, developing imaginal discs of *Drosophila* become mildly hypoxic as they grow towards their final size. A first hint came from transcriptomic analysis of precisely timed L3 discs, which suggests a progressive decrease in

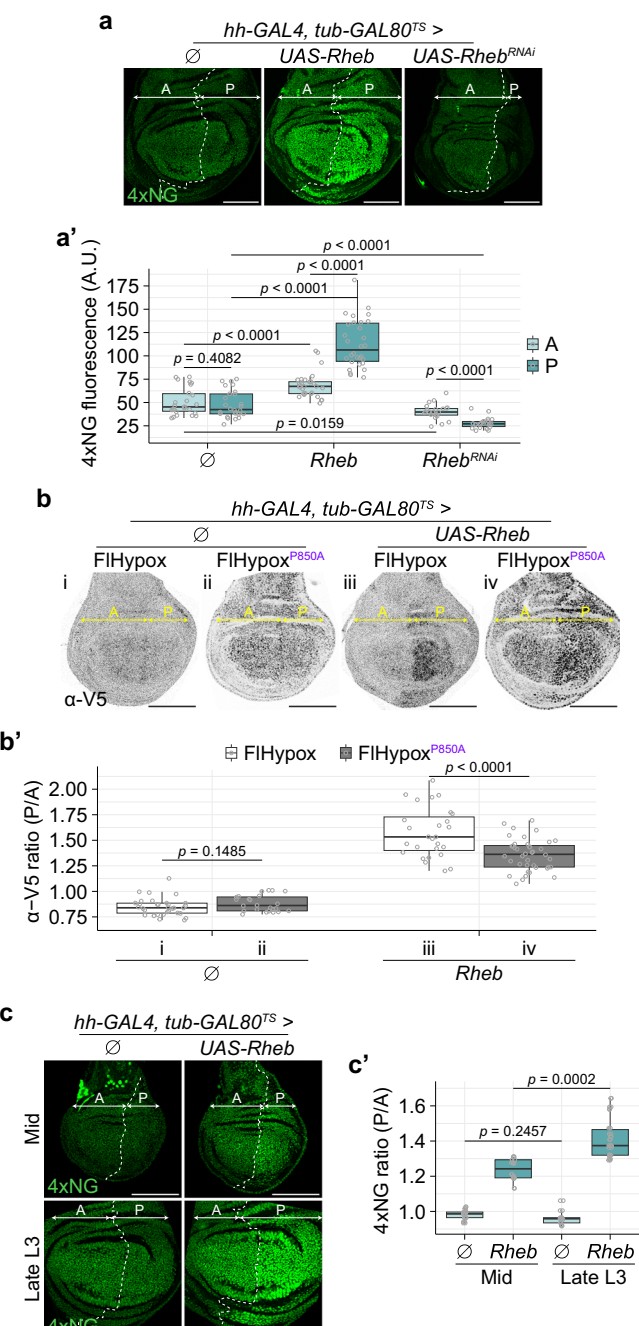

**Fig. 4 | Excess TOR activity leads to hypoxia in a tissue-size-dependent manner. a-a'** Fluorescence from SyHREns is altered by modulation of Rheb levels. Over-expression of Rheb in the P compartment (with *hh-GAL4*) enhances the SyHREns signal. Note that fluorescence intensity in the A compartment of Rheb-overexpressing discs is higher than in the A compartment of control discs lacking the *UAS-Rheb* transgene (∅), suggesting a non-autonomous effect on SyHREns activity. Knockdown of *Rheb* reduces SyHREns signal also both autonomously and non-autonomously. Representative images are shown in (**a**) (all discs imaged and processed under identical conditions, allowing intensities to be compared) and quantification of wing pouch fluorescence intensity is shown in (**a'**). For images shown here and in subsequent figures, the *hh-GAL4* expression domain (P com-partment) is recognised by the absence of Ci staining and indicated with a double-headed arrow. The A-P boundary was outlined manually (based on Ci staining) with a dotted white line. Scale bar = 100 µm. $n \geq 13$ discs for each genotype, except for data of *Rheb^RNAi^* where $n = 11$. After assessing normality of the data, statistical sig-nificance was assessed by a two-sided unpaired Wilcoxon signed-rank test. **b-b'** Representative images of discs stained with anti-V5 show that Rheb overexpression in the P compartment (with *hh-GAL4*) stimulates FlHypox significantly more than FlHypox^P850A^. For analysis of the fluorescence from the *ubi-mCherry* module from the same samples, see Supplementary Fig. 5a, b. Fluorescence in the P compart-ment was normalised to that in the A compartment by calculating the P/A ratio plotted in **b'**. Scale bar = 100 µm. $n \geq 14$ for each condition, except for data of FlHypox in ∅ where $n = 12$. Depending on the normality of the data, statistical significance was assessed by either a two-sided unpaired $t$ test or a two-sided unpaired Wilcoxon signed-rank test. **c-c'** Fluorescence from SyHREns increases upon Rheb overexpression in the P compartment. The fluorescence P/A ratio plotted in **c'** shows that this effect is more pronounced in late L3 discs than in mid-L3. Scale bar = 100 µm. $n \geq 10$ for each time point. After confirming normality of the data, statistical significance was assessed by a two-sided unpaired $t$ test.

OXPHOS, with a commensurate increase in glycolysis. This observation motivated us to develop FlHypox, a transgenic sensor of HIF-1α sta-bility and SyHREns, a transgenic reporter of HIF-1α activity. With these reporters and a commercial Hypoxyprobe™ assay, we confirmed our initial inference from our RNA-Seq data. This oxygen limitation likely imposes a constraint on growth. Growth requires ATP, which is much more efficiently produced in aerobic conditions than in the absence of oxygen[84]. Indeed, oxygen levels have been shown to constrain the final animal size[85,86]. As such, oxygen is analogous to a resource (like nutrients) in the logistic model that describes the evolution of popu-lation size under limited resources[14,87]. Therefore, it is conceivable that limited oxygen availability could account for growth deceleration in wing imaginal discs and perhaps in a wide range of biological tissues. It will nevertheless be difficult to establish a causal relationship, at least until a method is developed to clamp intracellular oxygen at specific levels.

Progressive depletion of oxygen in L3 imaginal discs suggests that oxygen supplies are limiting, not only in tumours as previously shown[88], but also during normal development. Animals rely exclusively on environmental oxygen. Before being absorbed by tissues and cells, oxygen is distributed throughout the body by the circulatory system in vertebrates or the tracheal system in insects. These systems can respond locally to oxygen deprivation (e.g., by stimulation of blood vessel formation[89] or terminal tracheal branches[90]), although global oxygen levels remain limited by the architecture of the main supply routes. Evidence from *Manduca sexta*, a holometabolous insect like *Drosophila*, shows that the tracheal system only expands between larval instars[91]. Therefore, during each instar, organismal oxygen sup-ply is unlikely to scale with individual growing tissues and organs. This could explain, in part, the rising level of hypoxia in L3 imaginal discs. Subsequent steps in the oxygen delivery system (e.g., transport across cell membranes) must also be limited since P compartment-specific Rheb-induced hypoxia does not readily equilibrate with the A com-partment (Fig. 5a). Overall, evidence suggests that both global and local oxygen supplies could be limiting.

Excess TOR/insulin signalling has been documented to boost the stability and nuclear localisation of HIF-1α[63,92–94]. There is also evidence that TOR signalling can directly control the transcriptional activity of HIF-1α independently of stabilisation in mouse monocyte cells[58]. However, the rise of FlHypox in response to Rheb overexpression shows that TOR activity also controls the level of HIF-1α protein. Two independent assays, with FlHypox and Hypoxyprobe™, suggest that this is mediated, at least in part, by hypoxia. The question arises, then, on how TOR signalling triggers hypoxia. In cancer cell lines, inhibition of the phosphatidylinositol 3-kinase (PI3K) and the mammalian TOR (mTOR) pathway reduces oxygen consumption by reducing mito-chondrial respiration coupled to ATP production[95]. It is therefore possible that during healthy and normal development, the demand for growth for ATP stimulates ATP production through the breakdown of nutrients by OXPHOS, an oxygen-intensive process. This would, in turn, dampen TOR signalling and perhaps at the same time trigger a progressive metabolic shift towards glycolysis, as our RNA-Seq analysis

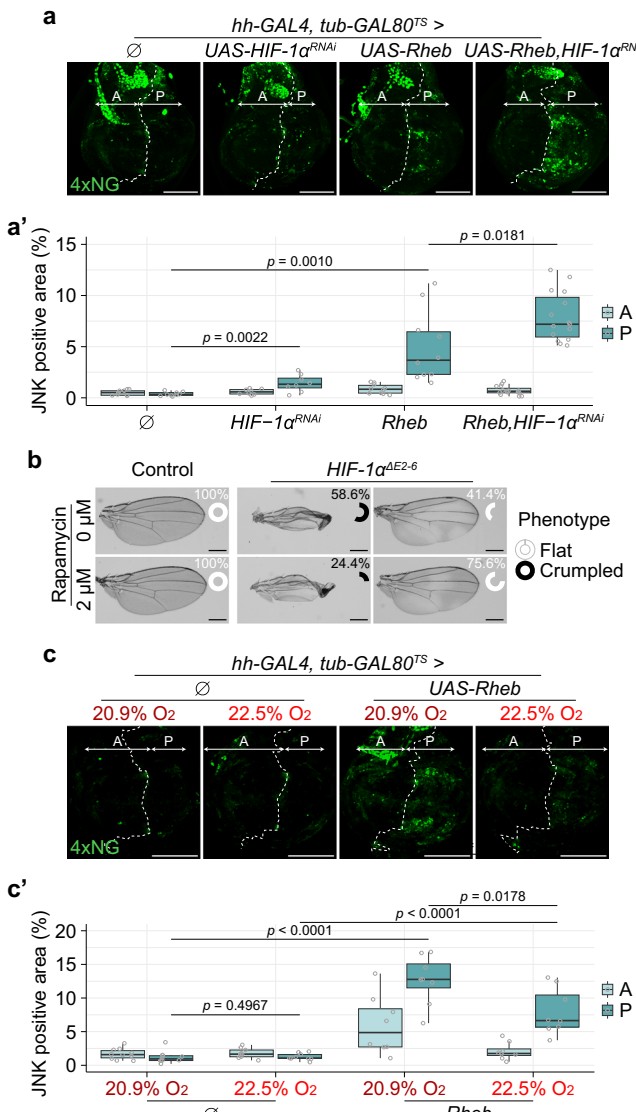

**Fig. 5 | HIF-1α reins in cellular stress induced by excess TOR activity. a-a'** HIF-1α limits tissue stress caused by TOR activity. Either *HIF-1α* knockdown or Rheb overexpression induces cellular stress, detected with the JNK reporter TRE-NLS4xNG. Knockdown of *HIF-1α* markedly exacerbates the stress-inducing effect of Rheb overexpression. Representative images are shown in (**a**) (all discs imaged under identical conditions), and quantification of wing pouch fluorescence area ratio is shown in (**a'**). Scale bar = 100 μm. $n \geq 9$ discs for each genotype, except for data of *HIF-1α^{RNAi}* where $n = 8$. After confirming normality of the data, statistical significance was assessed by a two-sided unpaired *t* test. **b** The 'crumpled' wing phenotype of *HIF-1α* KO flies is partially rescued by Rapamycin. Representative images and quantification of phenotype penetrance are shown. The penetrance of each phenotype is illustrated by an inset donut chart, with white indicating the proportion of flat wings. The scale bar represents the same length in all panels. $n \geq 150$ for the control wings and $n \geq 78$ for the *HIF-1α^{ΔE2-6}* homozygous wings. **c-c'** Stress induced by excess TOR signalling is alleviated by mild hyperoxia (22.5% $O_2$). Two right-hand panels show that the increase of TRE-NLS4xNG activity caused by Rheb overexpression in the P compartment at 20.9% is reduced to 22.5% $O_2$. Representative images are shown in (**c**) (all discs imaged under identical conditions), and quantification of wing pouch fluorescence area ratio is shown in (**c'**). Scale bar = 100 μm. $n \geq 8$ for each condition. Depending on the normality of the data, statistical significance was assessed by either a two-sided unpaired *t* test or a two-sided unpaired Wilcoxon signed-rank test.

suggests. According to this scenario, TOR signalling would put pressure on oxygen availability through boosting demand (Fig. 6). We cannot, however, exclude the possibility that TOR signalling could also affect supply (e.g., by reducing the cell surface-area-to-volume ratio and hence the efficiency of import across the plasma membrane). In summary, even though the underlying mechanism remains to be determined, TOR signalling puts pressure on oxygen supply during normal development, and this is the likely reason for HIF-1α-dependent dampening of TOR activity through transcriptional activation of *REDD1*[70–72] and a metabolic shift away from OXPHOS[51,52].

Our study highlights that growth exerts negative feedback on itself: TOR signalling activates HIF-1α, which in turn suppresses TOR signalling (Fig. 6). Artificially boosting TOR signalling leads to cellular stress, as indicated by activation of JNK signalling. A likely scenario is that excessive oxygen demand leads to a shortage and, hence, depletion of terminal electron acceptors for OXPHOS and production of ROS, which are known to damage DNA, proteins, and lipids, thus triggering cellular stress. As we have shown, such stress is exacerbated by *HIF-1α* knockdown and alleviated by a mild increase in environmental oxygen, indicating that excess TOR signalling causes cellular stress by putting impossible demands on available oxygen supply. *HIF-1α* knockdown in otherwise wild-type animals leads to mild cellular stress, suggesting that normal growth operates near the limit of available oxygen. Moreover, the wing phenotype of *HIF-1α* mutants is partially reverted by inhibition of TOR activity (i.e., reducing demand). Collectively, these observations suggest that the HIF-1α-mediated feedback loop ensures that during normal development, growth-induced oxygen consumption is not allowed to exceed available supply.

## Methods

### *Drosophila* strains and husbandry
*Drosophila* strains used in this study are summarised in Table 1. Flies were reared on fly food containing 5.5 g/L agar, 50 g/L glucose, 30 g/L wheat flour, 70 g/L yeast, 4.5 mL/L Propionic acid (Sigma-Aldrich, P5561), 19.5 mL/L Nipagin/Bavistin solution, 2.5 mL/L Penicillin/Streptomycin solution (Sigma-Aldrich, P4333) and ultrapure water. Unless otherwise noted, flies were maintained at 25 °C in a Sanyo incubator with 12-hour light/dark cycles.

### Generation of TRE-NLS4xNG
4xTPA response element (TRE)[96] was inserted upstream of a minimal heat shock (hs) promoter that drives the expression of a NeonGreen tetramer with a nuclear localisation signal (NLS4xNG)[20]. To enhance translation, a myosin heavy chain intervening sequence (IVS) and a synthetic AT-rich 21-bp sequence (Syn21) were placed upstream, and a highly-efficient p10 polyadenylation (polyA) signal was placed downstream of NLS4xNG. pJFRC81-10XUAS-IVS-Syn21-GFP-p10[69] was a gift from Gerald Rubin (Addgene plasmid #36432). The resulting TRE-NLS4xNG plasmid was injected into a fly line carrying an attP40 site by the Fly Facility of the Francis Crick Institute.

### Generation of FlHypox (Fly-Hypoxia) and FlHypox^{P850A}
To generate FlHypox, full-length *Drosophila HIF-1α/sima* cDNA (consisting of exons 1–12) tagged with a C-terminal viral protein-derived V5 epitope was cloned from pAc5.1/Sima[97] (a gift from Stefan Luschnig and Thomas Gorr). Three single amino acid substitutions (Q339E, V413E and Y417T) were subsequently introduced within the HIF-1β dimerisation domain to prevent dimerisation with HIF-1β, thereby inhibiting the activation of downstream hypoxia-related genes. A NeonGreen monomer (mNeonGreen) was inserted between amino acid residues S725 and A726, located within the ODD. This construct was then placed downstream of a *ubiquitin* (*ubi*) promoter and

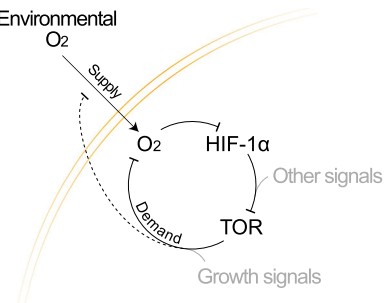

**Fig. 6 | HIF-1α-mediated negative feedback of TOR signalling onto itself.**
Environmental oxygen is transported into the organism before being imported through the cell membrane (gold lines). TOR signalling (which integrates multiple pro-growth signals) leads to reduced oxygen availability by increasing the demand for oxygen and/or possibly also affecting supply. The ensuing mild hypoxia activates HIF-1α, which in turn dampens TOR signalling, closing the negative feedback loop and preventing TOR from excessively depleting oxygen supplies.

**Table 1 | *Drosophila* strains**

| Genotype | Chromosome | Source |
|---|---|---|
| *w1118 iso31* | X | Ref. 109 |
| *Ldh-GFP* | III | Ref. 53 |
| *hsp70-GAL4, UASt-nlsTimer* | III | Ref. 56 |
| *ubi-EGFP-ODD, ubi-mRFP-nls* | III | Ref. 57 |
| *FlHypox* | III | This study |
| *FlHypox^P850A* | III | This study |
| *SyHREns* | II | This study |
| *SyHREns^mut* | II | This study |
| *vg-tub-GAL4* | II | Ref. 96 |
| *UAS-HIF-1α/sima^RNAi* | III | BDSC, 26207 |
| *UAS-REDD1/scyl^RNAi* | II | VDRC, GD25506 |
| *hh-GAL4, tub-GAL80^TS* | III | Lab stock |
| *UAS-HIF-1α/sima* | III | Ref. 110 |
| *UAS-Rheb* | II | BDSC, 9688 |
| *UAS-Rheb^RNAi-V20* | III | BDSC, 33966 |
| *TRE-NLS4xNG* | II | This study |
| *HIF-1α^>E2-6>* | III | This study |
| *βTub85D-Flp* | X | BDSC, 7196 |
| *HIF-1α^ΔE2-6* | III | This study |
| *HIF-1α/sima^KG07607* | III | Ref. 82 |
| *Df(3 R)BSC502* | III | BDSC, 25006 |

terminated with a p10 polyA signal. The same plasmid also included a second *ubi* promoter driving expression of mCherry, which also terminated with a p10 polyA signal. The resulting 2ubi-mCherry-FlHypox plasmid was injected into a fly line carrying an attP2 site by the Fly Facility of the Francis Crick Institute. Successful candidates were first identified by ubi-mCherry fluorescence and subsequently confirmed by PCR.

To generate FlHypox^P850A, a single amino acid substitution P850A was introduced into the 2ubi-mCherry-FlHypox plasmid. This mutation prevents hydroxylation of HIF-1α, thereby stabilising the protein by preventing its degradation. The resulting 2ubi-mCherry-FlHypox^P850A plasmid was injected into a fly line carrying an attP2 site by the Fly Facility of the Francis Crick Institute. Successful candidates were first identified by ubi-mCherry fluorescence and subsequently confirmed by PCR.

### Generation of SyHREns (Synthetic HRE nuclear sensor) and SyHREns^mut

To generate SyHREns, 5×HRE was synthesised using the GeneArt Custom Gene Synthesis Services (Thermo Fisher Scientific). It consists of five 81-bp repeats. Each repeat contains a HRE2, a HRE3 and the IVS, obtained from the 5′ upstream region of the sequence encoding *hph* isoform B that is upregulated in a HIF-1α-dependent manner in response to hypoxia[68]. Subsequently, 4×TRE in the plasmid of TRE-NLS4xNG was replaced by 5×HRE. The plasmid was injected into the BDSC, 9740 fly line by BestGene® to generate the SyHREns transgenic fly line.

To generate SyHREns^mut, a mutated version of 5×HRE (5×HREmut) was synthesised using the same service. This construct is identical in sequence to 5×HRE, except for a point mutation in each HRE2, in which the core HIF-1α binding motif (A/G)CGTG was altered to (A/G)AGTG, thereby disrupting HIF-1α binding[68]. The mutated sequence was used to replace the 5×HRE in the SyHREns plasmid. The final plasmid was also injected into the BDSC, #9740 fly line by BestGene® to generate the SyHREns^mut transgenic fly line.

### Generation of *HIF-1α^ΔE2-6* allele

A conditional allele *HIF-1α^>E2-6>* was initially created via CRISPR-Cas9 and homologous recombination-mediated repair as previously described[98]. Briefly, *HIF-1α* was made conditional by substituting the region spanning exon 2 to exon 6 with an identical sequence flanked by two flippase recognition target (FRT) sites. CRISPR target sites were selected in unconserved regions, with one located in the intronic region between exon 1 and exon 2 (GGAATAGCTGCCGAGAACTTTGG) and another in the intronic region following exon 6

(CGTTAGTTGGTGGGTTGCAATGG). To generate the rescuing construct pTVmCherry-HIF-1α^>E2-6>, both the 5′ homology arm and the exon 2 to exon 6 of *HIF-1α* flanked by two FRT sites (incorporated within the primers) were integrated upstream of a pax-mCherry selection marker through Gibson Assembly, whereas the 3′ homology arm was integrated downstream of the selection marker. The resulting pTVmCherry-HIF-1α^>E2-6> plasmid was co-injected with sgRNAs into embryos from the nanos-Cas9 line. This injection was performed by the Fly Facility of the Francis Crick Institute. Successful candidates were first identified by pax-mCherry fluorescence and subsequently confirmed by PCR. Following this, *β*-Tubulin at 85D (*β*Tub85D)-Flp (BDSC, 7196) was applied to excise the exon 2 to exon 6 in the male germline, thereby generating the *HIF-1α^ΔE2-6* allele.

### Quantification of wing disc volume

*Drosophila* larvae were selected at the L2-L3 transition (72 h AEL) and subsequently allowed to develop for specific periods before being inverted in phosphate buffered saline (PBS), fixed in 4% Methanol-free Pierce™ Formaldehyde (PFA) (Thermo Fisher Scientific, 28906) for 45 minutes (min) and stained with Invitrogen™ Alexa Fluor™ 647 Phalloidin (Thermo Fisher Scientific, A22287, concentration 3:800). To preserve the three-dimensional (3D) structure, the discs were dissected in PBS and mounted in 1% low melting point agar (Sigma-Aldrich, A9414) in PBS as previously described[26]. The mounted discs were immediately imaged with an upright Leica TCS SP5 confocal microscope using a ×20 immersion objective with a z step of 1 μm. Since phalloidin labels all the actin-containing cells, it stains the entire wing disc tissue. The staining allows for the 3D reconstruction of the tissue using Imaris (RRID:SCR_007370) and relatively accurate measurement of the tissue volume.

### Quantification of wing disc growth rate

A logistic regression model was employed to describe the correlation between the volumes (*V*) of wing discs and their developmental stages over time (*t*). This model is particularly suited to scenarios where growth is initially rapid but decelerates as it approaches a maximum value, mirroring the physiological growth processes. To estimate the

**Table 2 | Number of wing discs used for RNA extraction per replicate**

| Time point | Disc number |
|---|---|
| 80 h AEL | 100 |
| 88 h AEL | 45 |
| 96 h AEL | 25 |
| 104 h AEL | 20 |
| 112 h AEL | 16 |
| 116 h AEL | 12 |
| 118 h AEL | 12 |

parameters for the model, a self-starting logistic regression model was applied. This approach automatically creates initial values for the parameters based on the provided data without manual initialisation, and it is described by the following equation:

$$V = \frac{Asym}{1 + e^{\frac{tmid - t}{scal}}} \tag{1}$$

Where:
- $V$ denotes the volume of the wing discs at developmental time $t$,
- $Asym$ is the asymptotic maximum volume that the wing discs can achieve, analogous to the carrying capacity in traditional logistic models,
- $e$ is the base of the natural logarithm,
- $t_{mid}$ represents the time at which the volume is at half of its maximum value ($Asym/2$), serving as an inflection point in the context of growth,
- $scal$ is a scaling parameter that affects the steepness of the curve, influencing how quickly the volume increases towards the asymptotic maximum.

The rate of change of volume with respect to time was determined by calculating $dV/dt$, the first derivative of $V_t$. This provides insight into the speed of volume increase at different developmental time points. The growth rate at a given time point was determined by dividing $dV/dt$ by the instantaneous volume ($V_t$), quantifying the relative increase in volume as a percentage of the instantaneous volume:

$$Growth\ rate = \frac{dV/dt}{V_t} \tag{2}$$

This measures the efficiency of volume increase relative to the size of the wing disc at different developmental time points, providing a detailed view of growth dynamics. All analyses were conducted using RStudio[99].

**Bulk RNA-Seq sample preparation**

*Drosophila* larvae were selected at the L2-L3 transition (72 h AEL) and subsequently allowed to develop for specific periods before the wing discs were dissected in PBS on ice within 20 min. Discs were then snap-frozen using dry ice and ethanol. The number of discs used for a single replicate at each time point is shown in Table 2. Three biological replicates were collected at each time point.

After all the replicates were collected, RNA was extracted using the RNeasy Mini Kit (Qiagen, 74104), following the manufacturer's protocol. Sequencing libraries were then prepared using the KAPA mRNA HyperPrep Kit (Roche, KK8581), following the manufacturer's protocol. The Advanced Sequencing Facility of the Francis Crick Institute performed single-end 75 bp sequencing with 25 million reads per sample using a HiSeq 4000 System (Illumina).

**Bulk RNA-Seq analysis**

The raw reads were analysed and normalised by the Bioinformatics and Biostatistics team of the Francis Crick Institute. The raw reads were quantified using version 3.0 of NFCore's rna-seq pipeline[100], with starrsem as the alignment method against release 86 of Ensembl's BDGP6 genome. Subsequently, these read counts were analysed using Bioconductor's package DESeq2 version 1.30.1[101] - differential genes were identified between time points by using a negative binomial model including main effects for time (as a nominal variable) and experimental batch without any interaction, using Ward tests to assess the significance of shrunk[102] fold-changes between pairs of time points, using a false-discovery threshold of 1%[103].

Subsequently, to identify candidate genes potentially correlating with the decreasing growth rate, the Pearson correlation coefficient ($r$) between the changing transcript level over time and the decreasing growth rate was evaluated for each gene using RStudio[99]. Genes with $r > 0.8$ (decreasing transcript level over time) or $r < -0.8$ (increasing transcript level over time) were subjected to KEGG pathway analysis using the Database for Annotation, Visualisation and Integrated Discovery (DAVID)[104]. The identified KEGG pathway terms with a $p$ value < 0.05 were then subjected to graphic representation using RStudio[99].

For heatmap representation, transcript level at each time point was further normalised to the maximum value observed over time for each gene. A distance matrix was generated using Euclidean distance for continuous gene expression data. Subsequently, genes were clustered by assessing their similarities using the ward.D2 method[105]. All the analysis was conducted using RStudio[99].

**Larval oxygenation assay**

Larvae expressing *hs-GAL4 > UASt-nlsTimer* were selected at L2-L3 transition (72 h AEL). Both 72 and 84 h AEL larvae were simultaneously subjected to a heat shock treatment at 37 °C for 1 hour to induce a pulse of *nlsTimer* expression. Subsequently, nlsTimer was allowed to mature for 31 hours in vivo in both age groups of larvae. As described by Lehner and colleagues[56], the larvae were mounted in 90% glycerol in PBS and immobilised at −20 °C for 20 min before being imaged with an upright Leica TCS SP5 confocal microscope using a ×10 immersion objective with a z step of 1 μm.

**Validation of hypoxia sensors by hypoxia treatment**

Homozygous larvae were selected at the L2-L3 transition (72 h AEL) and allowed to develop for an additional 24 h to reach 96 h AEL. For each experiment, the larvae were transferred into vials (15 larvae per vial) containing 1 mL 0.5% 0.5% low-melting-point agarose (Sigma-Aldrich, A9414) in MilliQ deionised water. The vials were then evenly divided into two groups: the treatment group was placed in a hypoxia chamber set to a specific oxygen level in a Sanyo MIR-154 incubator at 25 °C; the control group was placed in a similar chamber within the same incubator but exposed to environmental normoxia (20.9% oxygen). Both chambers were humidified using wet paper towels. After the treatment, the larvae were inverted within 20 min and fixed for imaging. Oxygen level in the hypoxia chamber was precisely maintained by an oxygen controller (Coy Laboratory Products) connected to nitrogen and oxygen gas cylinders. The oxygen chambers were custom-built by the Alex Gould Lab and the Making Lab of the Francis Crick Institute.

For FlHypox and FlHypox^P850A, the hypoxia treatment was performed at 5.0% oxygen for 1 hour. Immediately after the treatment, the larvae were transferred onto ice prior to dissection.

For Hypoxyprobe, larvae were reared on fly food supplemented with 1 mg/mL Hypoxyprobe™ (Hypoxyprobe, Inc., HP-500mg) dissolved in MilliQ deionised water from 72 to 96 h AEL. The hypoxia treatment was performed at 5.0% oxygen for either 2.5 or 16 hours. Hypoxyprobe™ forms adducts with thiol-containing proteins in

**Table 3 | Antibodies**

| Antibody | Host | Source | Concentration |
|---|---|---|---|
| V5 | Mouse | Thermo Fisher Scientific, R960-25, RRID: AB_2556564 | 1:500 |
| PAb2627 | Rabbit | Hypoxyprobe, Inc., HP PAb2627 | 1:200 |
| pS6 | Rabbit | A gift from Jongkyeong Chung, Ref. 111 | 1:5000 |
| pS6 | Rabbit | A gift from Felipe Karam Teixeira, Ref. 112 | 1:500 |
| pS6 | Rabbit | This study | 1:100 |
| S6 (54D2) | Mouse | Cell Signalling Technology, 2317, RRID:AB_2238583 | 1:100 |
| Ci | Rat | DSHB, 2A1, RRID:AB_2109711 | 1:100 |
| β-Tubulin | Mouse | DSHB, E7, RRID:AB_528499 | 1:500 |
| Alexa Fluor™ Plus 647, anti-mouse | Goat | Thermo Fisher Scientific, A32728, RRID: AB_2633277 | 1:1000 |
| Alexa Fluor™ Plus 555, anti-rabbit | Goat | Thermo Fisher Scientific, A32732, RRID:AB_2633281 | 1:1000 |
| IRDye® 800CW anti-rabbit IgG | Donkey | LI-COR Biotechnology, 926-32213, RRID:AB_621848 | 1:10,000 |
| IRDye® 680RD anti-mouse IgG | Donkey | LI-COR Biotechnology, 926-68070, RRID:AB_10956588 | 1:10,000 |
| Alexa Fluor™ 647, anti-rat | Goat | Thermo Fisher Scientific, A-21247, RRID:AB_141778 | 1:1000 |

hypoxic cells, which are detected by the HP PAb2627 antibody (Hypoxyprobe, Inc., concentration 1:200, Data Sheet: https://site.hypoxyprobe.com/knowledge-center-articles/HP-PAb2627-Antibody-Insert.pdf).

For SyHREns, the hypoxia treatment was performed at a specific oxygen level (depending on the experiment) for 2.5 hours.

**Hypoxyprobe™ feeding during development under normoxia**
Larvae were selected at the L2-L3 transition (72 h AEL) and allowed to develop for an additional 20 h to reach 92 h AEL. Both 72 and 92 h AEL larvae were simultaneously reared on fly food supplemented with 2 mg/mL Hypoxyprobe™ (Hypoxyprobe, Inc., HP-500mg) dissolved in MilliQ deionised water for 24 h prior to dissection and fixation.

**Mathematical modelling**
For external oxygen levels between 5.0% and 18.0%, the average SyHREns intensities were normalised relative to their respective normoxia controls and plotted against the corresponding external oxygen levels ($O_2$). To quantify the correlation between normalised SyHREns intensity ($I$) and $O_2$ within this range, a linear regression model was applied, described by the following equation:

$$I = mO_2 + k \qquad (3)$$

Where:
- $I$ represents the normalised average SyHREns intensity,
- $m$ is the slope of the regression line, indicating the change in intensity with external oxygen levels ($O_2$),
- $O_2$ is the external oxygen level,
- $k$ is the y-intercept of the regression line.

In contrast, the average SyHREns intensities at 18.0% and 20.9% are not statistically significantly different, suggesting SyHREns do not respond to $O_2$ within this higher range. Therefore, for external oxygen levels between 18.0% and 20.9%, the average SyHREns intensities ($I_c$) were found to be independent of external oxygen levels ($O_2$). Consequently, a linear regression model with a slope of zero ($m = 0$) was applied, described by the following equation:

$$I_c = c \qquad (4)$$

Where:
- $I_c$ represents the constant average SyHREns intensity under 18.0% –20.9% $O_2$,
- $c$ is the y-intercept of the regression line, denoting the constant value of average SyHREns intensity within a higher range of $O_2$.

**Sensitivity of SyHREns**
The sensitivity of SyHREns, defined as the mildest hypoxia to which SyHREns can respond within the 2.5-hour exposure, was determined by identifying the intercept of the two linear regression functions. This intercept occurs at $O_2 = 17.09\%$, marking the transition point below which SyHREns begins to respond to decreasing oxygen levels.

**Dynamic range of SyHREns**
The dynamic range of SyHREns, which quantifies the extent of its response from the sensitivity threshold down to 5.0% $O_2$, was calculated as the fold-increase in intensity from 17.09% to 5.0% $O_2$. This analysis revealed that SyHREns demonstrates an approximately four-fold increase in intensity across its responsive oxygen range, indicating its capability to dynamically report changes in external oxygen level under specified experimental conditions and imaging with an upright Leica TCS SP5 confocal microscope.

**Immuno- and endogenous fluorescence analysis**
*Drosophila* larvae were inverted in PBS, fixed in 4% PFA for 30 min, washed in PBS for 10 min and then permeabilised in PBS with 0.1% Triton X-100 (0.1% PBST) for 30 min. For immunofluorescence imaging, the samples were blocked in 5% normal goat serum (NGS) (Thermo Fisher Scientific, 01-6201) in 0.1% PBST for 30 min before being incubated in primary antibodies diluted in 0.1% PBST at 4 °C overnight. Following three washes in 0.1% PBST for 15 min each, the samples were incubated in secondary antibodies diluted in 0.1% PBST at room temperature for 2 hours. After a further three 15-minute washes in 0.1% PBST, the samples were incubated in Vectashield® with DAPI (2BScientific, H-1200-10) at 4 °C overnight. Antibodies used in this study are shown in Table 3. For endogenous fluorescence imaging only, the samples were directly incubated in Vectashield® with DAPI at 4 °C overnight post-permeabilisation. Subsequently, the wing discs or eye discs were dissected in PBS, mounted in Vectashield® with DAPI, and imaged with either an upright Leica TCS SP5 or an inverted Leica Falcon SP8 confocal microscope using a ×40 immersion objective with a z step of 0.7 µm.

**Western blot analysis of wing discs**
Larvae were selected at the L2-L3 transition (72 hours AEL) and subsequently allowed to develop for specific periods before the wing discs were dissected in PBS on ice and lysed in 1× RIPA Lysis Buffer (Merck Millipore, 20-188) with 1× Ethylenediaminetetraacetic acid (EDTA)-Free Halt™ Protease Inhibitor Cocktail (Thermo Fisher Scientific, 87785). The lysates were then subjected to 15 cycles of sonication (20 seconds on, 20 seconds off) at a low frequency, using the Bioruptor® Plus (Diagenode, B01020001) at 4 °C. The total protein concentration in

each lysate was determined using the Pierce™ Bicinchoninic Acid (BCA) Protein Assay Kit (Thermo Fisher Scientific, 23227). Subsequently, the samples were denatured in a mixture of 1× Invitrogen™ Bolt™ LDS Sample Buffer (Thermo Fisher Scientific, B0007) and 0.25× Invitrogen™ Bolt™ Sample Reducing Agent (Thermo Fisher Scientific, B0009) at 95 °C for 5 min and cooled to room temperature. They were then loaded (4 μg total protein per well) alongside the PageRuler™ Plus Prestained NIR Protein Ladder (Thermo Fisher Scientific, 26619) onto the Invitrogen™ 4–12% Bolt™ Bis-Tris Plus Mini Protein Gels (Thermo Fisher Scientific, NW04120BOX, NW04122BOX or NW04125BOX) and subjected to electrophoresis using 1× Invitrogen™ MES SDS Running Better (Thermo Fisher Scientific, B0002). Following the electrophoresis, proteins were transferred from the gel onto the Immobilon®-FL PVDF Membrane (Merck Millipore, IPFL00010) in wet conditions at 120 Volts for 70 min using an Invitrogen™ Mini Gel Tank (Thermo Fisher Scientific, A25977). After that, the membranes were first blocked in 5% skimmed milk in PBS with 0.1% Tween-20 (0.1% PBSTw) for 30 min before being incubated in primary antibodies diluted in 0.1% PBSTw with 5% skimmed milk at 4 °C overnight. Following three washes in 0.1% PBSTw for 15 min each, the membranes were incubated in secondary antibodies diluted in 0.1% PBSTw with 5% skimmed milk and 0.01% sodium dodecyl sulphate (SDS) at room temperature for 2 hours. After a further three 15-minute washes in 0.1% PBSTw with 0.01% SDS, the membranes were imaged using an Odyssey CLx Imaging System (LI-COR). The protein band intensities were quantified using Image Studio™ (LI-COR). Antibodies used in this study are shown in Table 3.

## Antibody production

The anti-*Drosophila* phospho-RpS6 antibody was generated using the Polyclonal Antibody Development Services (Covalab) by immunising rabbits with the phospho-peptide, EAKRRR[pS]A[pS]IRE. The immunoaffinity column was prepared by coupling the phosphor-peptide to 1 mL of activated Sepharose beads. The antisera from the rabbits were loaded onto the peptide-sepharose column and incubated for 1 hour at 37 °C. After several washes of the column, the elution of bound antibody was performed using elution buffer containing 0.02% sodium azide. Fractions containing the antibody were pooled, and the final concentration of immunopurified antibody was determined by reading the optical density at 280 nm using UV spectrophotometer. The immunopurified antibody was then validated by ELISA and Western blot.

## O-propargyl-puromycin (OPP) incorporation assay

The protein synthesis rate was determined using the Click-iT™ Plus OPP Alexa Fluor™ 594 Protein Synthesis Assay Kit (Thermo Fisher Scientific, C10457) as previously described[106]. Briefly, larvae were selected at the L2-L3 transition (72 h AEL) and subsequently allowed to develop for specific periods before being inverted in Gibco™ Schneider's *Drosophila* Medium (Thermo Fisher Scientific, 21720024) at room temperature within 15 min. The samples were then incubated with 1 μM OPP (Supplementary Fig. 5c) in Gibco™ Schneider's *Drosophila* Medium at room temperature for 15 min before being washed in PBS and fixed in 4% PFA. Following a wash in PBS, a 30-min permeabilisation in 0.1% PBST and a 30-min blocking in 5% NGS in 0.1% PBST, the samples were subsequently incubated with the Click-iT™ reaction cocktail in the dark at room temperature for 30 min. After being washed once with the Click-iT™ Reaction Rinse Buffer, the samples were incubated in Vectashield® with DAPI. The wing discs were then dissected in PBS, mounted in Vectashield® with DAPI, and imaged with either an upright Leica TCS SP5 or an inverted Leica Falcon SP8 confocal microscope using a ×40 immersion objective with a z step of 0.7 μm.

## Hybridisation chain reaction (HCR) and image acquisition

The visualisation of mRNA in wing discs was performed using the HCR™ IF Bundle (Molecular Instruments), following a protocol adapted from Patel and colleagues[107]. The initial preparation involved inverting larvae in PBS, fixing them in 4% PFA for 1 hour, followed by a 10-min wash in PBS and a 30-min permeabilisation in 0.1% PBSTw. Subsequently, the samples were pre-hybridised in 300 μL pre-warmed Probe Hybridisation Buffer at 37 °C for 30 min, followed by an overnight incubation in 300 μL Probe Hybridisation Buffer containing 1 pmol of each probe set at 37 °C. Both steps employed a thermoshaker set at 300 rpm. Post-hybridisation, the samples were subjected to four 15-min washes in 1 mL pre-warmed Probe Wash Buffer at 37 °C, and then two 5-min washes in 600 μL 5× SSCT at room temperature. 100 μL Amplification Buffer containing 6 pmol of each hairpin was incubated at 95 °C for 90 seconds before being cooled to room temperature in the dark. After the washes, the samples were pre-amplified in 400 μL pre-warmed Amplification Buffer at room temperature for 30 min, followed by an overnight incubation in 100 μL hairpin-containing Amplification Buffer at room temperature in the dark. Both steps employed a thermoshaker set at 300 rpm. Post-amplification, the samples were subjected to two 5-min and two 30-min washes in 400 μL 5× SSCT at room temperature, followed by three 10-min washes in PBS. After that, the samples were incubated in Vectashield® with DAPI at 4 °C overnight before the wing discs were dissected in PBS, mounted in Vectashield® with DAPI, and imaged with an inverted Leica Falcon SP8 confocal microscope using a ×40 immersion objective with a z step of 0.7 μm.

## Compartmental genetic perturbation

To achieve compartmental genetic perturbation in wing discs, *hedgehog*(*hh*)-*GAL4* was used together with the temperature-sensitive *tubulin*(*tub*)-*GAL80*^TS. The larvae were reared at 18 °C, the *GAL80*^TS permissive temperature, for 5 days AEL. To activate the compartmental genetic perturbation, the larvae were transferred to 29 °C, the *GAL80*^TS restrictive temperature. At this temperature, *GAL80*^TS is inhibited, allowing the *GAL4*-driven gene expression. Following two days at 29 °C, the larvae were inverted and fixed for imaging.

## Compartmental TOR overactivation under hypoxia or hyperoxia

For compartmental TOR overactivation under hypoxia or hyperoxia conditions, the larvae were initially reared at 18 °C for five days AEL on fly food under environmental normoxia. The treatment group was then transferred to the hypoxia or hyperoxia chamber set to a specific oxygen level at 29 °C. Meanwhile, the control group was transferred to a similar chamber at 29 °C but kept under environmental normoxia. After two days of *GAL80*^TS inactivation under either hypoxia or hyperoxia, the larvae were inverted and fixed for imaging.

## Quantification of fluorescence intensity

For wing disc fluorescence signals, quantification of intensity was specifically focused on the wing pouch, distinguishable by the folds surrounding this region. The A-P boundary within the wing pouch was identified by anti-Ci staining. Using Fiji, both the wing pouch and the A-P boundary were outlined manually, and a threshold excluding the background signals was applied to create a binary mask. The resulting regions of interest (ROIs) were used to determine the average fluorescence intensity either across the entire wing pouch or within each compartment. Unless otherwise noted, a single confocal plane that optimally captures the entire wing pouch was used for quantification and image representation.

For nuclear fluorescent signals, thresholding was applied to isolate the nuclei as a binary mask. The resulting ROIs were used to determine the average fluorescence intensity across the entire wing pouch, within each compartment, or throughout the larva.

## Quantification of fluorescence area

Using Fiji, both the wing pouch and the A-P boundary were outlined manually, and the area of each compartment was determined. To

**Article**

segment the fluorescence-positive regions, thresholding was applied to isolate these regions as a binary mask. The resulting ROIs were used to determine the total area within each compartment. The fluorescence area was calculated as a percentage of the compartment area occupied by the fluorescence-positive regions.

## Rapamycin treatment

Embryos were laid either on fly food containing 2 µM Rapamycin (Merck Supelco, 37094) dissolved in ethanol or on fly food containing an equivalent volume of ethanol alone (0 µM Rapamycin). Following hatching, the larvae were allowed to feed and develop into adult flies before the penetrance of wing crumpling was quantified.

## Sample preparation and image acquisition of adult wings

Adult flies were incubated in a solution comprising 70% glycerol (v/v) and 30% ethanol (v/v) at room temperature overnight before being washed in distilled water. Wings were dissected in isopropanol (Thermo Fisher Scientific), mounted in Euparal (Agar Scientific), and dried at room temperature for 24 hours. Images were acquired using a Zeiss Axioplan 2 microscope with a Leica MC190 HD camera, using a ×2.5 objective for general wing morphology or a ×10 objective for detailed views of the trichomes.

## Quantification of adult wing trichome density

For assessment of trichome density, a systematic approach was applied where four equally sized ROIs were selected within the wing, specifically between the longitudinal veins L2 and L3, L3 and L4, L4 and L5, and L5 and L6 using Fiji. In these ROIs, a threshold exclusive for the trichome sockets, excluding the background signals, was applied to create a binary mask. The density of trichomes was calculated based on the average count of trichome sockets across the four ROIs for each wing, presented as the number of trichomes per unit area.

## Timing of larval development

Embryos were laid on agar plates supplemented with yeast paste over 4-hour intervals from 8:00 to 20:00. L1 larvae were collected at 24 hours AEL from the agar plates and transferred into vials containing fly food. Each vial housed 35 larvae, with either six or nine vials prepared per genotype, depending on the vitality of the fly lines ($n = 210$ or $n = 315$). Three days after the transfer, the number of larvae that had pupariated was recorded every 4-hour from 8:00 to 20:00 for 3–5 days. The proportion of larvae that transitioned to pupae within each population was then plotted against developmental time, generating a curve of larval developmental timing. The average timing of larval-pupal transition is defined by the time point where 50% of the population becomes pupae[108].

## Reporting summary

Further information on research design is available in the Nature Portfolio Reporting Summary linked to this article.

## Data availability

The RNA-Seq data generated in this study have been deposited in the Gene Expression Omnibus database under accession code GSE264119 https://www.ncbi.nlm.nih.gov/geo/query/acc.cgi?acc=GSE264119. An interactive table of the RNA-Seq data, featuring individual expression plots for each gene in the developing L3 wing disc, is available at https://rnaseq-wingdisc-zhaoetal.shinyapps.io/zhaoetal2025/. Access is limited to a maximum 25 hours per month for all users. FlyBase ID and names for the genes identified by RNA-Seq are available at FlyBase (http://flybase.org/). Source data are provided with this paper. All data supporting the findings of this study are available within the article or from the corresponding authors upon reasonable request. Source data are provided with this paper.

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

## Acknowledgements

We thank Ana Bolhaqueiro for the characterisation of the *TRE-NLS4xNG* strain. The Light Microscopy STP at the Francis Crick Institute provided assistance with imaging, and the Genomics STP prepared libraries from mRNA and performed the sequencing (Deb Jackson, Jimena Perez-Lloret, Marg Crawford, and Robert Goldstone). Ben Nicholls-Mindlin offered advice on bootstrapping the relative growth rate and provided comments on the manuscript. Y.Z. was the recipient of a PhD studentship from the Francis Crick Institute. The initial phase of this work was funded in part by an EMBO fellowship to GPM (ALTF 238-2018) and a Wellcome Trust Investigator Award to J.P.V. (206341/Z/17/Z). The project was completed with core funding from the Francis Crick Institute to J.P.V. (FC001204). The Francis Crick Institute receives its core funding from Cancer Research UK, the UK Medical Research Council, and the Wellcome Trust.

## Author contributions

This project was conceived by G.P.M., J.P.V and Y.Z. C.A. built the FlHypox$^{(P850A)}$, the TRE-NLS4xNG and the *HIF-1α$^{>E2-6>}$* strains. G.K. performed differential gene expression analysis from the raw RNA-Seq reads. Y.Z. executed most of the experiments and data analysis, including bioinformatics. Data interpretation was done collectively by all authors. Y.Z. wrote the first draft, which was subsequently modified and edited by all the authors.

## Funding

## Competing interests

The authors declare no competing interests.
