## [Transparent Peer Review file · Nature Communications]

HIF-1 α -mediated feedback prevents TOR signalling from depleting oxygen supply and triggering stress during normal development

Corresponding Author: Dr Jean-Paul Vincent

Version 0:

Reviewer comments:

Reviewer #1

(Remarks to the Author)

The authors have nicely addressed the concerns raised in the original review. The manuscript has a large amount of solid, interesting data - findings which will certainly be of interest to a broad audience studying organ size, growth, oncogenic signaling and hypoxia.

Reviewer #2

(Remarks to the Author)

The authors have fully addressed my concerns and provided important clarifications to the points raised. The novelty of the work is more clearly presented and I appreciate the thoughtful and considered responses. In my view, the work is suitable for publication.

Reviewer #3

(Remarks to the Author)

In this report, Zhao et al investigate the signalling pathways behind development, in particular in wing disks between L1 and L3. They document a growth reduction and identify genes in an unbiased manner related to oxidative phosphorylation, biomass production and glycolysis. They go on to determine the role of HIF (SIMA) and Tor in this pathway. Again all of these concepts are not new, but the precise developmental stage analysis is a valuable new aspect. The HIF probe is a great new tool as well. Most of my comments were addressed but the aspect of ROS is not covered well in the discussion. The addition of the low resolution hypoxia probe is a good one.
